# Design of Far-Infrared High-Efficiency Polarization-Independent Retroreflective Metasurfaces

**DOI:** 10.3390/mi15040538

**Published:** 2024-04-17

**Authors:** Siliang Zhou, Siyu Dong, Tao He, Jingyuan Zhu, Zhanshan Wang, Xinbin Cheng

**Affiliations:** 1MOE Key Laboratory of Advanced Micro-Structured Materials, Shanghai 200092, China; siliang_zhou@tongji.edu.cn (S.Z.); dongsy@tongji.edu.cn (S.D.); hetao@tongji.edu.cn (T.H.); wangzs@tongji.edu.cn (Z.W.); chengxb@tongji.edu.cn (X.C.); 2Institute of Precision Optical Engineering, School of Physics Science and Engineering, Tongji University, Shanghai 200092, China; 3Shanghai Frontiers Science Center of Digital Optics, Shanghai 200092, China; 4Shanghai Institute of Intelligent Science and Technology, Tongji University, Shanghai 200092, China

**Keywords:** far-infrared, retroreflection, polarization-independent, metasurface, high-efficiency, RCWA

## Abstract

Retroreflective gratings serve as fundamental optical elements in nanophotonics, with polarization-independent diffraction efficiency being one of the critical parameters for assessing their performance. In the far-infrared spectral range, traditional retroreflective gratings typically refer to metal echelette gratings, but their diffraction efficiency cannot approach 100% due to metal absorption. In the visible and near-infrared spectral ranges, metal echelette gratings have gradually been replaced by all-dielectric metasurfaces because dielectric materials exhibit negligible absorption at specific wavelengths. However, there is still a lack of relevant research in the far-infrared range, mainly due to the weak control capability of the existing devices over the polarization-independent phase. Here, we propose a kind of all-dielectric retroreflective metasurface composed of asymmetric pillars and freely tunable aperiodic multilayer films. The pillar structure can achieve polarization insensitivity, and the insufficient modulation capability of the dielectric materials can be compensated for by aperiodic Ge/ZnS films. The designed metasurface achieves the diffraction efficiency by RCWA, with the maximum larger than 99% and the overall reaching 95% (9.3–9.6 µm). We have provided detailed explanations of the design methodology and fabrication process. Our work lays the groundwork for further exploration and application of far-infrared lasers.

## 1. Introduction

As fundamental optical elements, gratings are commonly used in optical systems such as information communication [1], pulse compression [2], spectral analysis [3,4], laser beam combining [5,6,7,8,9], and various other functions [10,11]. Retroreflection is one of the functions of gratings, referring to the ability to reflect light back along the incident direction. The most common type of retroreflective grating is metal echelette grating [12], which utilizes the periodic structure of the grating and diffraction phenomena to selectively reflect light of specific wavelengths through the blaze condition. Retroreflectors find applications in various fields, such as atmospheric research [13], optical communications [14], and optical metrology [15]. Due to the characteristics of infrared lasers, especially in the far-infrared range, which enable long-distance transmission in the atmosphere and selective absorption for different substances, they offer unprecedented advantages in both the research and industrial sectors. Polarization-independent diffraction efficiency is a crucial parameter for evaluating the performance of retroreflective gratings. Achieving efficiency close to 100% over a broadband range for both transverse electric (TE) and transverse magnetic (TM) polarizations can effectively enhance the system’s output power and serve as the basis for implementing other complex functionalities. However, the efficiency of the most common gold-coating echelette gratings still cannot exceed 95% due to the limitations imposed by metal absorption [16,17]. Different types of retroreflective diffractive elements are being researched to enhance efficiency.

Metasurfaces, as a type of two-dimensional metamaterial, possess powerful capabilities for controlling phase, amplitude, and polarization [18,19]. They play a significant role in various fields [20,21,22,23], including anomalous reflection [24,25]. In recent years, considerable efforts have been made to improve the efficiency of reflective metasurfaces [26,27]. In the microwave band, there have been studies reporting a retroreflective metasurface using subwavelength-thick reconfigurable C-shaped resonators, achieving phase modulation under different polarizations [28]. However, in the optical band, the efficiency sharply declines due to the high absorption of metal structures, especially at large deflection angles [29]. To address this issue, researchers are increasingly exploring the use of dielectrics instead of metals. For instance, Lalanne demonstrated an ultra-broadband diffractive metasurface in the infrared spectrum using alternating aperture structures [30]. However, its large period of 140 µm, containing 50 subunits, severely affects the angular resolution, and higher-order diffraction losses hinder the efficiency improvements. Dong [31] proposed a metagrating comprising an all-dielectric Bragg reflector and a periodic metasurface with freeform-shaped dielectric resonators, enabling broadband polarization-independent perfect Littrow diffraction at optical frequencies. Qi [32] introduced a kind of high-efficiency polarization-insensitive retroreflective metagrating with cascaded nano-optical modes on high-reflection films, doubling the manufacturing tolerance. Li [33] proposed a free-space optical link exploiting a flat retroreflector based on a metasurface doublet. Arbabi [34] fabricated a planar monolithic near-infrared retroreflector composed of two layers of silicon nanoposts, which reflects light along its incident direction with a normal incidence efficiency of 78% and a large halfpower field of view of 60°. Although polarization-independent retroreflectors in the visible and near-infrared spectrum have been extensively researched, achieving polarization-independent and highly efficient retroreflectors working at the far-infrared range remains a significant bottleneck. Due to the reliance on the effective refractive index for modulation, dielectric materials are more sensitive to changes in wavelength, especially in the far-infrared wavelength range. This results in limitations to the control capability of an all-dielectric metasurface.

Here, we propose a metasurface design with high polarization-independent efficiency. By combining pillar structures with aperiodic multilayer films, we effectively enhance the diffraction efficiency for both TE and TM polarizations simultaneously. The asymmetric pillars achieve polarization insensitivity, while the aperiodic multilayer films enhance the phase modulation capabilities longitudinally, addressing the issue of the insufficient control capability of the effective refractive index by the upper structure. Through rigorous coupled-wave analysis (RCWA) simulations, the designed metasurface achieves a polarization-independent diffraction efficiency with the maximum exceeding 99% and an overall efficiency of 95% around 9.5 µm.

## 2. Theory and Simulation Result

The designed metasurface consists of an upper periodic structure and lower multilayer films. As shown in Figure 1, when light passes through the upper periodic structure, its energy is distributed across different diffraction orders. The −1st order reflection angle is equal to the incident angle when retroreflecting [35]. In this case, apart from the reflection orders, there will also be transmitted energy passing through the upper structure. The role of the lower multilayer films is to provide sufficient reflectance to couple all the transmitted energy passing through the upper structure into the reflection orders. Each layer of dielectric in the multilayer films serves as a basic reflection unit, capable of enhancing the reflection for a specific wavelength of light. By arranging multiple basic reflection units and setting different refractive indices or thicknesses for each layer unit, it is possible to selectively adjust which wavelength of light is enhanced in the reflection. When there are many layers and the reflection wavelength variation between different units is small, such a structure becomes a Bragg mirror that can strongly reflect over a continuous wavelength range. Alternating layers of Ge/ZnS films with thicknesses of λ/4*n* (where *n* is the refractive index Ge/ZnS = 4.2/2.2) can form a Bragg mirror in the far-infrared wavelength range, achieving 100% reflectance.

### 2.1. Design Principles of Polarization-Independent Retroreflective Metasurface

The grating structure induces diffraction, causing incident light to be reflected at different angles. Higher-order diffraction reduces the efficiency of the required order. Therefore, to maximize the reflection efficiency of the −1st order, we need to avoid the generation of higher-order diffraction, which requires adjusting the period of metasurface and the incident angle. According to the grating formula,
(1)$$D(n_{i}\sin\theta_{i}+n_{r}\sin\theta_{r})=m\lambda ,$$
When retroreflection occurs in air, the equation can be simplified to
(2)2Dsinθ=mλ,
where *D* is the period of metasurface (the same width of subunits in the x and y directions), θ is the incident angle, *m* is diffraction order, and λ is central wavelength. It could be obtained that, in order to limit the diffraction order to 0th and −1st, a smaller period and a larger incident angle are necessary conditions. In this configuration, the incident light energy exists only in the 0th and −1st channels after being reflected from the metasurface.

According to the law of energy conservation, suppressing the energy of the 0th order reflection is an indirect means to enhance the efficiency of the −1st order [36]. After analysis, it is determined that the energy of the 0th order reflection consists of two components: direct reflection and coupled transmission. When their amplitudes are equal and phases are opposite (a phase difference of π), destructive interference occurs, leading to energy cancellation, which can be written as
(3)$$\Delta \Phi =\arg\left(U_{r}^{0}\right)-\arg\left(U_{t}^{0}\right)=\pi .$$$U_{r}^{0}$ and $U_{t}^{0}$ are the amplitudes of the 0th order reflection and transmission, respectively. At this point, the energy of the 0th order reflection can be effectively suppressed. The amplitude is related to the transmittance and reflectance, and can be adjusted through the upper periodic structure. Each subunit arranged periodically can be viewed as a waveguide structure, where electromagnetic waves passing through the subunit accumulate phase. The phase is mainly composed of the transmission phase $\Phi_{t}$ and the phase at the interface junction $\Phi_{c}$, which can be written as [37]
(4)$$\Phi_{t}=2k_{0}h_{s}n_{eff},$$
(5)$$\Phi_{c}=\varphi_{0}+\varphi_{c},$$
where $k_{0}$ = 2π/λ is the wave vector. The transmission phase $\Phi_{t}$ is determined by the modal effective refractive index $n_{eff}$ and the height of the structure $h_{s}$. $\Phi_{c}$ is determined by the phase at the interface between the grating and air $\phi_{0}$ and the reflection phase of the multilayer films $\phi_{c}$. Therefore, the phase difference can be simplified as
(6)$$\Delta \Phi =\Delta \Phi_{t}+\Delta \Phi_{c}=2k_{0}h_{s}\Delta n_{eff}+\Delta \varphi_{0}+\Delta \varphi_{c}.$$
As seen from the equation, the magnitude of the phase difference can be controlled by the parameters of the upper periodic structure and the reflection phase of the multilayer films.

During the design of the metasurface, the parameters of the upper structure are fixed first. We choose 2D rectangular pillars because of their insensitivity to polarization [38]. The fill factor of the periodic structure affects the effective indices of Bloch modes for different diffracted propagating waves. By computing the difference in Bloch-mode effective indices $\Delta n_{e}ff$ for different fill factors in the x and y directions, their coverage range can be determined as shown in Figure 2a. It demonstrates that the coverage range of 2D rectangles is significantly larger than that of 1D gratings. To ensure high polarization-independent efficiency, $\Delta n_{e}ff$ between TM and TE polarizations needs to be close, as in
(7)$$\Delta n_{eff}\left(TM\right)=\Delta n_{eff}\left(TE\right)\;$$

In the process of selecting suitable structural parameters to meet the above requirements, there are two challenges worth noting. The first is that, since our metasurface operates in the far-infrared spectrum, smaller wave vectors make it difficult to achieve an appropriate phase difference, which typically translates to larger fill factors and heights for traditional structures. The second point is that achieving equal control of the phase under different polarizations is relatively challenging because subtle changes in the fill factors can cause simultaneous variations in both. However, we can control $\Phi_{c}$ to compensate for and adjust the phase difference ΔΦ with the designed metasurface after selecting appropriate parameters and determining $\Phi_{t}$. For periodic multilayer films, $\Phi_{c}$ is determined by the refractive index of the medium and polarization (close to 0 in this study). It is constrained due to its limited variability range. Nevertheless, aperiodic multilayer films provide the possibility to alter the reflection phase. By altering the thickness of the top two layers of the multilayer films, we found that $\Phi_{c}$ can cover −π to π, as shown in Figure 2b. This is helpful for controlling the phase difference because $\Phi_{c}$ excited by the aperiodic films is the same for TE and TM polarizations, which allows for simultaneous increases or decreases in the phase difference ΔΦ under different polarizations, greatly alleviating the design difficulty of the upper structure. Without reducing the reflectance, the aperiodic films effectively increased the range of phase modulation.

Materials commonly used for infrared applications include Si, Ge, oxides, fluorides, and sulfides. However, most of these materials exhibit absorption losses in the far-infrared wavelength range. Ge is chosen as the material for the upper periodic structure to avoid losses. The lower multilayer films not only need to avoid losses but also require a large refractive index difference between the high- and low-dielectric materials to achieve broadband high reflectance. We choose Ge and ZnS as the high- and low-refractive-index materials, respectively, because previous studies have shown that this combination offers high reflectance and low absorption in the far-infrared wavelength range, and it is feasible for fabrication [39].

In summary, by adjusting the parameters of the upper structure and the thickness of aperiodic films, the energy of the 0th order reflection of the metasurface can be effectively suppressed. Due to the polarization insensitivity of 2D rectangular pillars, the diffraction efficiency of the −1st order can reach 100% for both TE and TM polarizations.

### 2.2. Simulation Results of Designed Metasurface

The 2D simulation results of the designed metasurface are obtained by RCWA [40,41]. To achieve a scenario where only the 0th and −1st diffraction orders exist, the period of the metasurface *D* is fixed to be 5 µm, and the incident angle is fixed to be around 70°. After determining the initial solution through matched phases, we utilized Particle Swarm Optimization (PSO) [42] to obtain the final parameters, as shown in Figure 3a, including the width and height of the rectangular pillars, as well as the thickness of the aperiodic multilayer films. The Bragg mirror consists of seven layers of Ge/ZnS multilayer films, with their thickness determined by the refractive indices of the materials.

When the incident light passes through the grating structure, diffraction occurs. The diffracted lights of different orders reach the underlying multilayer films at different angles of incidence and undergo multiple scattering within the spacer. Therefore, we require underlying multilayer films with high reflectance. As shown in Figure 3b, the reflectance of the multilayer films consisting of Bragg mirrors and aperiodic films maintains 100% at different incident angles, which means that aperiodic films do not decrease high reflectance.

We computed the diffraction efficiency of the designed metasurface under TE and TM incidences using RCWA, as depicted in Figure 4a. In the far-infrared spectrum around 9.5 µm, the −1st order diffraction efficiency under both the TE and TM incidences can exceed 99%, with an average efficiency over 95% within a 300 nm bandwidth. According to the grating formula, when the −1st order diffraction efficiency approaches 100%, retroreflection is induced, meaning both the reflection angle and the incident angle are 70°. We validated the simulation results using the lateral electric field distribution plots. As shown in Figure 4b, the normalized electric field distribution is uniform. Retroreflection is achieved under 70° oblique incidence, which can be demonstrated by the direction of the electric field. The electric field direction is perpendicular to the direction of light propagation, making a 20° angle with the normal direction. This indicates that the metasurface concentrates almost all the energy into the −1st order, achieving perfect reflection.

The primary objective of this paper is to enhance the polarization-independent efficiency within a narrowband spectrum. When the wavelength of incident light deviates from the central wavelength, the reflected light exhibits slight angular deflection. In practice, for incident light with a wavelength of 9.5 µm, precise retroreflection requires oblique incidence at 71.8° according to grating Equation (Equation 2). We analyzed the variation in the angle between the reflection and incident angles within the design bandwidth as the wavelength changed, as shown in Figure 5a. The blue line represents the deflection angle, while the red line represents the average efficiency. The results indicate that the deflection angle remains within 10°. Furthermore, although our design fixes the incident angle at 70°, we also analyzed the deviation in the reflected light relative to the incident light when the incident angle varied within a certain range. As shown in Figure 5b, owing to a large incident angle and the high line density of the metasurface, when the incident angle varies within ±5°, the deflection angle remains within 10°. The diffraction efficiency at different angles exceeds 97%.

Since there is scant research on retroreflectors in the far-infrared range, we have chosen retroreflectors from different spectra for comparison. Our study focuses more on reflection efficiency, so we have compiled the following Table 1:

Currently, research on retroreflectors mainly focuses on small-angle working conditions near the normal. While retroreflection can be induced at various angles, peak efficiency is only achieved near normal incidence. Efficiency rapidly decreases with angle deviation. Through comparison, it can be observed that our designed metasurface operates with high polarization-independent efficiency (above 95%) within a certain range under oblique incidence (70°). Both the peak and narrowband efficiencies are significantly higher than in the existing works. This work opens up new possibilities for far-infrared laser applications.

## 3. Discussion

The challenge in designing far-infrared polarization-independent devices lies in the fact that the modulation capability is limited, making it difficult for the phase variation to meet the requirements for different conditions. However, our designed metasurface breaks through this limitation by employing phase-matching design principles. The design principle in Section 2.1 for phase difference is a necessary condition for achieving perfect diffraction. This is because we overlooked the influence of the higher-order Bloch modes and evanescent waves during the design process, so the phase matching and perfect diffraction are not entirely equivalent. In fact, the final structural parameters in Section 2.2 were obtained by reverse design, namely through over 800 iterations of Particle Swarm Optimization (PSO). However, PSO requires an excellent initial solution and variable range to avoid local optima or non-convergence issues, which is precisely why we conducted the reflection phase calculations first.

In theory, the modulation range of the rectangular pillars and the two layers of aperiodic film is limited. However, by altering the pillar shapes and increasing the number of aperiodic film layers, the degrees of freedom can be enhanced. Research has already utilized topological structures to replace pillar structures in design, further enhancing their modulation capability [31]. This can further increase the bandwidth of polarization-independent high efficiency, albeit with the risk of sacrificing peak efficiency.

Although metasurfaces have achieved many complex functions, there is still a long way before they can be practically applied. In our design process, we paid extra attention to the limitations encountered by the metasurfaces in practical applications, such as error analysis and fabrication feasibility and tolerance. The subsequent sections will provide detailed explanations regarding the simulation and fabrication methods.

## 4. Methods

### 4.1. Simulation Method

An open-source rigorous coupled-wave analysis (RCWA) solver, “RETICOLO” [43], was used to perform the full-wave simulation, including the calculation of diffraction efficiency, reflection phase, and electromagnetic field. The simulation of the two-dimensional structure was achieved by stacking layers with the same period in both the x and y directions to address diffraction issues. The (x, y) plane and the z-direction are, respectively, referred to as the lateral plane and the longitudinal direction. Initially, we defined the top layer (air) and the bottom layer (substrate). The incident light propagates along the z-axis direction from top to bottom. The z-axis direction is from bottom to top. The 2D structure has a period of 5 µm in both x and y directions, with refractive indices of 4.2 for Ge, 2.2 for ZnS, and 3.47 for Si. The system employs a plane wave with TE and TM polarization incident at 70°. The central wavelength is 9.5 µm. When calculating the effective refractive index coverage range, the scanning interval for the 1D grating is set to 0.01, while, for the 2D rectangular structure, it is set to 0.05.

The structural parameters are obtained through particle swarm optimization (PSO), with the average polarization-independent efficiency as the objective value. It is a parameter-light and relatively simple optimization method. Compared to genetic algorithms, it converges faster for high-dimensional optimization problems. However, PSO is prone to becoming trapped in local optima, thus requiring careful initialization. Firstly, a smaller learning factor can increase the thoroughness of the search but may lead to convergence to local optima. Conversely, a larger learning factor can expand the search range but might overlook the global optimum. Secondly, setting initial solutions based on problem characteristics or utilizing prior knowledge of the problem can mitigate this issue. Based on the phase theory calculations in Section 2.1 and considering fabrication feasibility, the fill factor of the pillar structure is in the range of 0.2–0.8 and the height of the pillar structure is in the range of 2–2.5 µm. The learning factor is set to 2 and the number of iterations exceeds 800.

### 4.2. Fabrication Method

In order to validate the theoretical accuracy and the feasibility of fabrication, we employed the Monte Carlo method to calculate the tolerance of design parameters [44]. We randomly sampled the design parameters within a tolerance range of ±10% for 500 iterations and obtained the average polarization-independent efficiency. By statistical analysis, we obtained the probability distribution of efficiency, as shown in Figure 6. The results indicate that the designed metasurface device exhibits excellent fabrication tolerance, with over 97% high efficiency achievable within the ±10% tolerance range.

Although the fabrication of the designed metasurface has not been carried out, consideration has been afforded to the feasibility of the fabrication methods. The fabrication process primarily includes physical vapor deposition (PVD), e-beam lithography (EBL), and ICP-RIE etching. The major fabrication steps of the metasurface are shown in Figure 7.

The Ge/ZnS multilayer films were initially deposited on a Si substrate through electron beam evaporation. Following the deposition of 7 sets of periodic reflection layers and 2 aperiodic layers, Ge was further deposited on top of them to serve as the grating layer. Subsequently, photoresist was spin-coated, serving as a mask. Following this, the pattern was written using electron beam lithography. After exposure, the resist was developed and fixed.

Furthermore, the fabrication of the upper structure also requires Ge ICP etching. Currently, our existing etching processes involve the preparation of pillar structures using CF_4_ and SF_6_ for Si and certain oxides. However, the Ge etching requires HBr and Cl_2_ as reactive chemicals [45,46], which presents a challenge for us. Following the etching process, any remaining photoresist will be removed using oxygen plasma. In summary, after the etching issue is resolved, the retroreflective metasurface can be successfully fabricated.

## 5. Conclusions

In this paper, we have proposed a novel design of a high polarization-independent diffraction efficiency retroreflective metasurface operating in the far-infrared spectrum theoretically. The quasi-3D all-dielectric metasurface consists of an upper layer of 2D rectangular pillars and a lower layer of aperiodic multilayer films. The rectangular pillars exhibit polarization insensitivity, ensuring that the transmission phases are close for both TM and TE incident light, which is a necessary condition for enhancing the polarization-independent efficiency. The aperiodic multilayer films can freely adjust the reflection phase, not only compensating for the deficiency in the transmission phase but also maintaining a phase difference of π, effectively suppressing the energy of the 0th order specular reflection. By combining the two-dimensional pillars with the aperiodic multilayer films, we achieved a maximum efficiency of over 99% for the −1st order, with the efficiency values reaching 95% in the entire spectrum range (9.3 µm–9.6 µm).

In summary, the retroreflective metasurface proposed in this paper can effectively replace the traditional metallic gratings in the far-infrared wavelength range due to its high polarization-independent efficiency and the high damage threshold of dielectric materials. It can perfectly reflect obliquely incident light back to its original direction without any energy loss. Micron-sized pillar structures with an aspect ratio close to 1:1.4 are also suitable for fabrication using e-beam lithography and ICP-RIE etching. This work provides significant advancements for applications such as far-infrared laser radar, spectral analysis, holographic imaging, and beyond. Future efforts will focus on expanding the bandwidth, the metasurface fabrication of far-infrared materials, and experimental testing.

## Figures and Tables

**Figure 1 micromachines-15-00538-f001:**
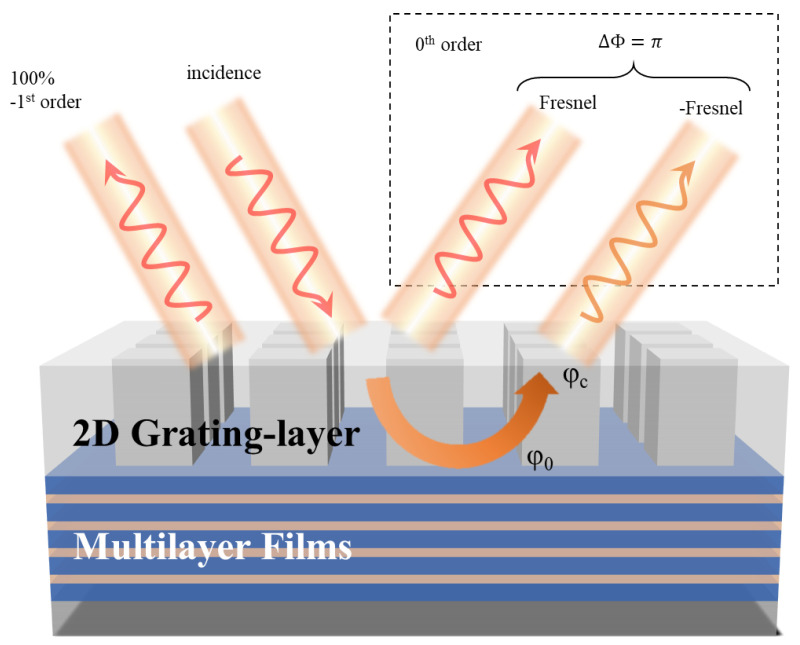
The diagram of metasurface system consisting of an upper grating layer and lower multilayer films. The −1st order reflection angle is equal to the incident angle when retroreflecting. The maximum of the −1st order efficiency can be obtained when the energy of other orders is suppressed.

**Figure 2 micromachines-15-00538-f002:**
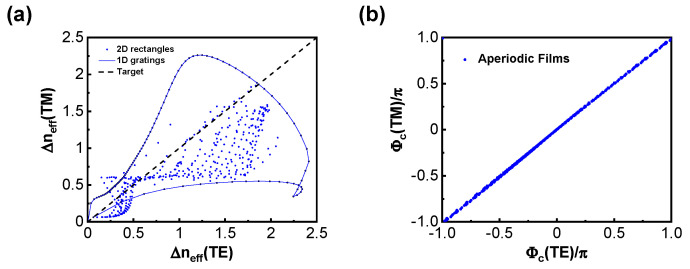
Phase modulation of the designed metasurface. (**a**) The coverage range of TE and TM effective index differences for pillars at different fill factors. The solid line represents the coverage range of 1D gratings, dots represent the coverage range of 2D rectangles, and the dashed line represents the target value. (**b**) The modulation range of the interface phase $\Phi_{c}$ of the aperiodic films under different polarization incidences (TE and TM). $\Phi_{c}$ is very close under different polarizations.

**Figure 3 micromachines-15-00538-f003:**
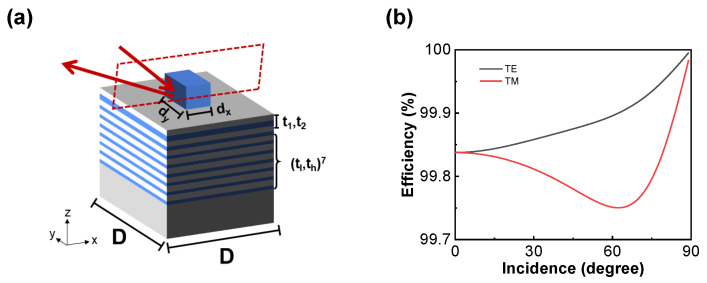
(**a**) Schematic diagram of the individual subunit. The widths of pillars are $d_{x}$ = 3.15 µm and $d_{y}$ = 3.55 µm. The height of pillars is h= 2.5 µm. The thickness values of aperiodic films are $t_{1}$ = 789 nm and $t_{2}$ = 753 nm. The thickness values of Bragg mirror are $t_{l}$ = 1080 nm and $t_{h}$ = 565 nm. (**b**) The reflectance of the multilayer films at different angles of incidence is close to 100%.

**Figure 4 micromachines-15-00538-f004:**
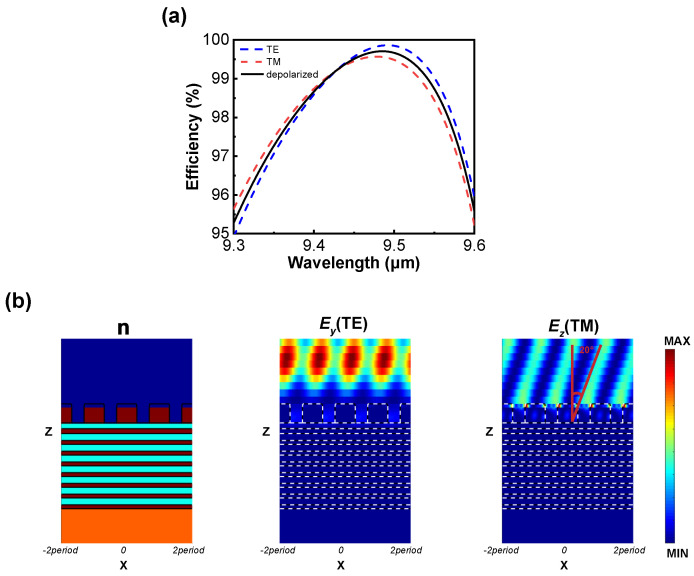
Simulation results of the designed metasurface. (**a**) Efficiency curve of the metasurface operating from 9.3 µm to 9.6 µm. (**b**) Electric field distribution profile for four periods under TE incidence (E*_y_*) and TM incidence (E*_z_*). From left to right are the refractive index distribution plot of the structure, the y-component of the electric field under TE incidence, and the z-component of the electric field under TM incidence.

**Figure 5 micromachines-15-00538-f005:**
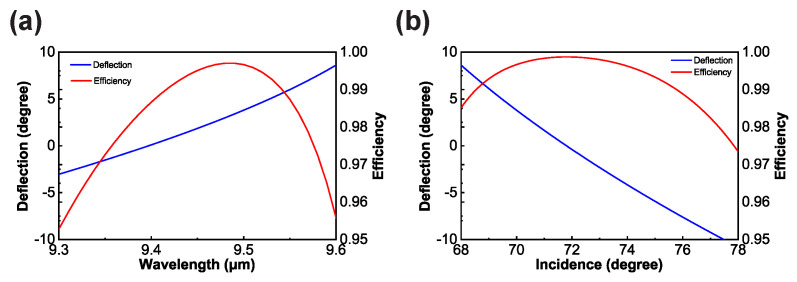
The deflection and average efficiency with (**a**) wavelength variation and (**b**) incidence variation.

**Figure 6 micromachines-15-00538-f006:**
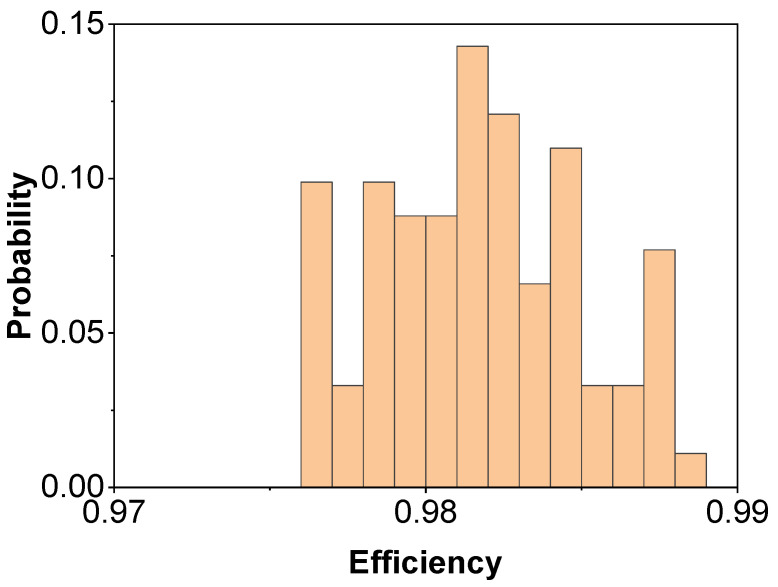
Fabrication tolerance simulations. Efficiency probability with tolerance (<±10%) for different parameters including $d_{x}$, $d_{y}$, *h*, $t_{1}$, and $t_{2}$.

**Figure 7 micromachines-15-00538-f007:**
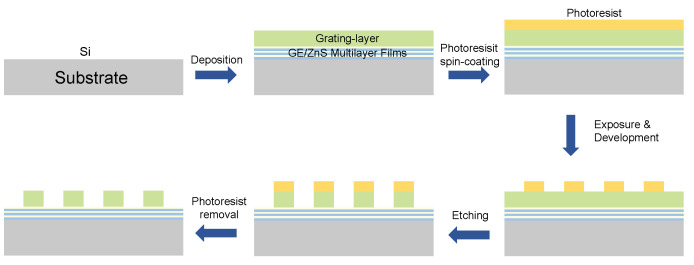
Fabrication process of the proposed device.

**Table 1 micromachines-15-00538-t001:** The characteristics of retroreflection metasurfaces from different studies, including incident angle, wavelength, and efficiency.

Research	Incident Angle	Wavelength	Efficiency
RCRs [28]	10°∼20°	20 mm	80∼90%
FRR [33]	0°∼25°	1550 nm	<96.8%
Planar Monolithic Retroreflector [34]	0°∼50°	850 nm	<78%
Our work	68°∼78°	9.5 µm	>95%

## Data Availability

Data underlying the results presented in this paper are not publicly available at this time but may be made available by the authors upon reasonable request.

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
