# Peer review of "Design of Far-Infrared High-Efficiency Polarization-Independent Retroreflective Metasurfaces"

_micromachines, 2024, doi:10.3390/mi15040538_

Round 1

Reviewer 1 Report

Comments and Suggestions for Authors

In this paper, the authors proposed a kind of all-dielectric reflective gratings that can achieve high working efficiency based on asymmetric pillars and aperiodic multilayer films. The simulated results indicate that the diffraction efficiency can reach up to 99%. Although this work seems interesting, there are still some problems need to be further illustrated.

1.     The authors named their devices as retroreflective metasurfaces, however, they do not show the retroreflection performance of the designed devices. In fact, previous works such as 10.1038/nphoton.2017.96 for retroreflective metasurfaces can reflect light along its incident direction with a certain incident range. I doubted that whether the proposed work can achieve such functionality.

2.     I suggest the authors use the term “polarization-independent” instead of “depolarization”.

3.     The dimensions of the pillars are miss labelled, “dx = 3.15m and dy = 3.55m. The height of pillars is h = 2.5m.”

4.     I suggest the authors add some fabrication tolerance simulations.

5.     In part 2.1, the authors show that the reflection phases are crucial for the perfect diffraction. I am wondering whether such theory can guide the design for the proposed device? It seems that the final results are still based on numerical simulations.

6.     When considering 2D metagratings in the far infrared, the following references may be helpful. 10.29026/oea.2023.220073, 10.1002/adma.202008157 and 10.1021/acsphotonics.8b00434.

Comments on the Quality of English Language

Minor editing of English language required.

Reviewer 2 Report

Comments and Suggestions for Authors

In the manuscript, the authors have designed a metasurface retroreflector optimized for far-infrared applications. It features asymmetric pillars and aperiodic multilayer films engineered to enhance diffraction efficiency and enable polarization insensitivity. Despite its merits, there are key aspects that necessitate further refinement. Crucially, the author's motivation, innovativeness, and practicality are not well described. Given these considerations, my recommendation at this juncture is major revision.

Questions:

1.        The paper could benefit from a more quantitative analysis of the device, specifically regarding the working angular range of the Metasurfaces. Can the authors provide such metrics?

2.        On page 3, line 73, the term 'period of metasurface' is used. What is being referred to here? And the 5 microns stated on line 131, from where to where is this measured?

3.        In lines 136-137, it's mentioned that the Bragg mirror achieves 100% reflectance at varying incident angles. Could the authors expound on the retroreflector's performance within this angle range?

4.        In lines 143-144, it is claimed, "the reflection angle matches the set incident angle." How was this conclusion reached? The authors are requested to include a more detailed quantitative analysis.

5.        What is the intended function of the “discussion” section in this article? What is the intended function of the “discussion” section in this article? The content inside seems to be a summary of the main text content.

6.        Additional details on how the aperiodic multilayer films compensate for the insufficient refractive index in the upper-layer dielectric materials would be valuable.

7.        The manuscript specifically highlights the metasurface's design for the far-infrared range but fails to address the imperative of retroreflectors at these wavelengths in the introduction.

8.        When discussing the generation of “high-efficiency retroreflective metasurfaces”, the inclusion of a table for comparison and summary relative to other studies would be beneficial.

Others:

1.        Figure 1 does not sufficiently illustrate the structure, and could potentially mislead readers; Furthermore, the clarity of figures, such as 4b, is inadequate for proper evaluation.

2.        The authors' introduction of metasurface retroreflectors is limited. It's advisable to elaborate on this topic and include more references to existing literature, for example:   https://doi.org/10.1038/nphoton.2017.96

https://onlinelibrary.wiley.com/doi/10.1002/adma.201802721

https://onlinelibrary.wiley.com/doi/full/10.1002/adom.202100796

3.        On page 1, line 23, the first letter in 'depolarization' requires capitalization.

4.        On page 3, line 69, the first letters of 'principles' and 'depolarization' need capitalization.

5.        It is recommended that the content undergo revision by a native English-speaker.

Round 2

Reviewer 1 Report

Comments and Suggestions for Authors

The authors have addressed all my concerns and this paper can be published in its current form.

Reviewer 2 Report

Comments and Suggestions for Authors

The authors have properly responded to my comments. Despite the incorrect format of Ref. 35, I believe the manuscript can be accepted in its current form.